# Druggable Biomarkers Altered in Clear Cell Renal Cell Carcinoma: Strategy for the Development of Mechanism-Based Combination Therapy

**DOI:** 10.3390/ijms24020902

**Published:** 2023-01-04

**Authors:** Youcef M. Rustum, Ryan Reis, Tara M. Rustum

**Affiliations:** 1Department of Internal Medicine, Carver College of Medicine, University of Iowa, Iowa City, IA 52242, USA; 2Department of Cancer Biology, Roswell Park Comprehensive Cancer Center, Buffalo, NY 14203, USA; 3Department of Dermatology, College of Medicine Medical Scientist Training Program, University of Iowa Carver, Iowa City, IA 52242, USA; 4Department of Dermatology, University of Iowa Hospitals and Clinics, Iowa City, IA 52242, USA; 5Roswell Park Cancer Institute, University of Pittsburgh, Pittsburgh, PA 15260, USA

**Keywords:** selenium, clear cell renal cell carcinoma, druggable targets, therapeutics potential

## Abstract

Targeted therapeutics made significant advances in the treatment of patients with advanced clear cell renal cell carcinoma (ccRCC). Resistance and serious adverse events associated with standard therapy of patients with advanced ccRCC highlight the need to identify alternative ‘druggable’ targets to those currently under clinical development. Although the Von Hippel-Lindau (VHL) and Polybromo1 (PBRM1) tumor-suppressor genes are the two most frequently mutated genes and represent the hallmark of the ccRCC phenotype, stable expression of hypoxia-inducible factor-1α/2α (HIFs), microRNAs-210 and -155 (miR_S_), transforming growth factor-beta (TGF-ß), nuclear factor erythroid 2-related factor 2 (Nrf2), and thymidine phosphorylase (TP) are targets overexpressed in the majority of ccRCC tumors. Collectively, these altered biomarkers are highly interactive and are considered master regulators of processes implicated in increased tumor angiogenesis, metastasis, drug resistance, and immune evasion. In recognition of the therapeutic potential of the indicated biomarkers, considerable efforts are underway to develop therapeutically effective and selective inhibitors of individual targets. It was demonstrated that HIF_S_, miR_S_, Nrf2, and TGF-ß are targeted by a defined dose and schedule of a specific type of selenium-containing molecules, seleno-L-methionine (SLM) and methylselenocystein (MSC). Collectively, the demonstrated pleiotropic effects of selenium were associated with the normalization of tumor vasculature, and enhanced drug delivery and distribution to tumor tissue, resulting in enhanced efficacy of multiple chemotherapeutic drugs and biologically targeted molecules. Higher selenium doses than those used in clinical prevention trials inhibit multiple targets altered in ccRCC tumors, which could offer the potential for the development of a new and novel therapeutic modality for cancer patients with similar selenium target expression. Better understanding of the underlying mechanisms of selenium modulation of specific targets altered in ccRCC could potentially have a significant impact on the development of a more efficacious and selective mechanism-based combination for the treatment of patients with cancer.

## 1. Introduction

Over the last decade, there has been a significantly expanded knowledge base of the molecular, immunological, and biological profile of tumor cells and their adjacent microenvironment, as well as of the mechanisms of altered expression of specific targets implicated in increased tumor angiogenesis, metastasis, and drug resistance. The knowledge gained has provided the rationale for the development of new drugs and therapies for cancer patients. Specifically, the development of targeted therapeutics, including immune checkpoint, multi-targeted tyrosine kinase, and mTOR inhibitors, revolutionized the treatment of patients with clear cell renal cell carcinoma (ccRCC) [1,2]. Durable and sustained responses were documented in approximately 30% of treated patients. However, the documented innate and/or acquired resistance to these drugs in the majority of ccRCC tumors highlights the need to identify and therapeutically evaluate alternative druggable targets to those under clinical evaluation. Identification of new druggable targets that are altered in the majority of ccRCC tumors could provide the basis for the development of mechanism-based treatment combinations that may further enhance the response to standard targeted therapeutics. ccRCC tumors are molecularly and immunologically heterogeneous tumors that express several dysregulated targets (Figure 1, Table 1). These include the following:(1)Biomarkers associated with unstable and leaky tumor vasculature [3,4,5].(2)Overexpression of the mutant Von Hippel-Lindau gene [6,7] and the Polybromo-1 (PBRM-1) tumor-suppressor gene [8,9,10].(3)The high incidence and stable expression of the hypoxia-inducible factor-1α and -2α (HIF_S_) proteins [11,12,13,14,15,16] and the upregulated oncogenic miR_S_-210/-155 [17,18,19,20,21,22,23].(4)The altered expression of mitochondrial lipid-metabolizing enzymes, carnitine palmitoyl transferase-1A and Perilipin-2 [24,25,26], enzymes regulated by the cooperative interaction of HIFs and oncogenic miRs.
ijms-24-00902-t001_Table 1Table 1Reported expression levels of biomarkers in ccRCC tumors with and without sarcomatoid differentiation.ReferenceMarkersSarc.ccRCCNno-Sarc ccRCC[27]PD-L1/TIL90%62%[27]PD-L154%17%[27]PD1/PD-L150%3%[11]HIF183%67%[11]HIF1/250% not expressed32%[28]miRs-210/-155over expressedover expressed[29]Nrf2not reported78%

Randomized phase III clinical trials of immune checkpoint inhibitor-based combination therapy have recently demonstrated superiority of the combination therapy over monotherapy in naïve patients with metastatic ccRCC (Figure 2, Table 2). Moreover, ccRCC with sarcomatoid features are more sensitive to immune checkpoint inhibitors than to VEGF(R) inhibitors, and express higher levels of the programmed death ligand-I (PD-L1) protein. Resistance and grade 3 and 4 toxicities, however, continue to represent major clinical challenges, and highlight the need to identify additional druggable targets. Although ccRCC tumors are molecularly and immunologically heterogeneous, the focus of this minireview is to identify additional biomarkers altered in ccRCC other than VEGF and PD-L1 that can be effectively downregulated by nontoxic doses of selenium. This minireview is not intended to provide a comprehensive review of ‘druggable ’targets altered in ccRCC. The focus, rather, is on specific targets, HIF_S_, miR_S_, and Nrf2, that are overexpressed in ccRCC tumors where selenium was determined an effective modulator. With the knowledge that ccRCC tumors are considered unresponsive to anticancer cytotoxic drugs that are active in other malignancies, along with the need to identify cytotoxic drugs that can potentially further enhance the antitumor activity of biologically targeted molecules, the overexpressed thymidine phosphorylase in ccRCC is being introduced as a target for the activation of 5-Fluorouracil pro-drug. The hope is that this minireview will stimulate preclinical and clinical research to confirm the therapeutic potential of targeting specific types of biomarkers altered in ccRCC by a defined dose and schedule of specific types of selenium molecules.

## 2. Clear Cell Renal Cell Carcinoma

Clear cell renal cell carcinoma (ccRCC) represents over 75% of RCC, is highly immunogenic, and is molecularly, immunologically, and histologically heterogeneous. Although mutation or inactivation of the Von Hippel-Lindau (VHL) tumor-suppressor gene is considered the hallmark of ccRCC, the second most mutated gene after VHL is PBRM1 [6,7,8,9,10]. Histologically, 5–15% of ccRCC express sarcomatoid features, which hold a unique molecular and immunological profile. While non-sarcomatoid ccRCC is generally considered chemotherapy-resistant, efficacy to specific types of chemotherapeutic agents alone and in combination with targeted molecules demonstrated some degree of efficacy with dose-limiting toxicity. Four phase III trials have been completed that evaluated the combination of VEGF/VEGFR-targeted therapy and immune checkpoint inhibitors (biologic molecules) vs. targeted therapy alone, namely with sunitinib, and results have been collected in untreated sarcomatoid and non-sarcomatoid ccRCC [1] (Table 2). For the 1753 patients without sarcomatoid differentiation treated with the combination of biologic-targeted molecules, the reported intention-to-treat (ITT) responses were as follows: overall response rate (ORR), complete response rate (CR), median progression-free survival (mPFS), and median overall survival (OS) were 47.4%, 6.3%, 12.1 months, and 34.8 months, respectively. For the 1756 patients without sarcomatoid differentiation treated with sunitinib alone, the reported ORR, CR, mPFS, and mOS were 30.9%, 1.8%, 9.1 months, and 30.8 months, respectively. For the 226 patients with sarcomatoid differentiation treated with the combination of biologic molecules, the ORR, CR, mPFS, and mOS were 52.8%, 11.1%, 7.9 months, and 24.8 months, respectively. For the 241 patients with sarcomatoid differentiation treated with sunitinib alone, the reported ORR, CR, mPFS, and mOS were 21.5%, 1.4%, 5.7 months, and 14.3 months, respectively. Collectively, the data in Figure 2 and Table 2 indicate that both with and without sarcomatoid differentiation exhibited higher response rates with the combination of biologic-targeted molecules than with sunitinib alone (Figure 2). For patients with ccRCC treated with axitinib alone in second-line therapy, the reported ORR, mPFS, and mOS were 18%, 8 months, and 18 months, respectively [30,31].

Regarding the expression patterns seen in ccRCC, research has shown that, while PD-L1 is expressed in 69/98 (70.4%) total ccRCC, PD-L1 is only expressed in 11.2% of cells with the wild Von Hippel-Lindau (VHL) tumor-suppressor gene [32]. Next-generation sequencing techniques have advanced our understanding of the biologic and molecular profile of ccRCC with identified loss of chromosome 3p and mutations of VHL and PBMR1. Sarcomatoid-ccRCC is associated with TP53-, PTEN-, and RELN-mutated genes, as well as TGF-β, while ccRCC is associated with VHL- and PBRM1-mutated genes [33,34,35]. ccRCC tumors with wild expression of VHL are characterized by a high-frequency presence of sarcomatoid features, chromosomal instability, and aggressiveness. It is well-documented that the tumor microenvironment and associated vasculature are unstable, leaky, and likely contributors to the instability of tumor vasculature, decreased delivery of drugs to tumor cells, and resistance of tumor cells to standard therapies [3,36]. The crosstalk between tumor vasculature and tumor tissues not only contributes to tumor growth and proliferation but also limits the delivery of drug concentrations sufficient for inhibition of tumor growth and metastasis. Thus, modulation of biomarkers that promote normalized tumor vasculature may offer the opportunity for the development of new, novel, and selective therapies for cancer patients.

## 3. Druggable Targets Altered in ccRCC

### 3.1. MicroRNAs-210/-155 (miR_s_) and Hypoxia-Inducible Factor-1α and -2α (HIF_S_)

Although many solid tumors express multiple tumor-suppressor and tumor-promoter microRNAs (miR_S_), the focus is on miR_S_-210 and miR-155, which are ubiquitously overexpressed in many solid tumors [37,38,39,40] and on the demonstration that these miRs are selenium targets in ccRCC [18]. miRs-210/-155 have been reported to regulate the expression of VHL [40], PBRM1 [9,10], HIFs, PD-L1, multidrug-resistant proteins [41,42], nuclear factor erythroid 2-related factor 2 (Nrf2) forkhead box p3 (FoxP3) transcription factors [43], and key mitochondrial lipid-metabolizing enzymes [44,45]. miR-210 and -155 are induced by hypoxia and considered hypoxia biomarkers, and these miRs are significantly overexpressed in normoxic ccRCC cell lines expressing HIFs and mutated VHL [17,46,47,48,49,50,51,52]. miRs also function as epigenetic modulators and as modifiers of DNA methylation. The oncogenic miRs regulate gene expression at the transcriptional levels by binding to the 3′ untranslated region (UTR) of the protein-coding target mRNA [50], thus resulting in translational repression and degradation. The 3′UTR of PD-L1 contains two binding sites for the oncogenic miR-155 to bind, resulting in the inhibition of translation of PD-L1 mRNA. Further, the promoter region of oncogenic miRs contains a functional hypoxia-responsive element to which HIFs and their regulated genes bind [50]. Functionally, HIF-1α is reported to regulate glycolysis, which is predominately involved in early stages of ccRCC development, while HIF-2α regulates genes associated with lipoprotein metabolism and is predominately involved in late stages [13]. Se-methylselenocysteine (MSC) and seleno-L-methionine (SLM) (selenium) were identified as highly effective inhibitors of constitutively expressed and HIFs [52,53,54]. Inhibition of HIFs by selenium is prolyl hydroxylase 2 (PHD2)- and proteasome degradation-dependent, independent of VHL status. The effects were associated with significant enhancement of the antitumor activity of anticancer therapeutics [55,56,57].

Histologically, ccRCC tumors are characterized by large and lipid-rich cytoplasmic deposits, and high vasculature density. HIFs and miRs are known to regulate the metabolic enzymes associated with the accumulation of these lipid droplets. Published data have demonstrated that the oncogenic miRs interact with carnitine palmitoyl transferase 1A and Perilipin 2 [24,25] key enzymes that regulate the cytoplasmic accumulation of lipid droplets with succinate dehydrogenase (SDH) and glycerol-3-phosphate dehydrogenase 1-like, which target prolyl hydroxylase (Figure 1). In tumors with mutated SDH or fumarate hydratase, the accumulated levels of succinate and fumarate inhibit prolyl hydroxylase, resulting in decreased PHD-dependent hydroxylation of HIFs, along with stable expression of HIFs [58,59,60,61]. Downregulation of GPD1L by miR_S_ and HIF_S_ in ccRCC tumors has been attributed to the inhibition of PHD-dependent hydroxylation. The overexpressed miR-210 was reported to inhibit GPD-1L, resulting in stable expression of HIFs. Although considerable efforts are underway to develop inhibitors of miRs and HIF_s_, in vivo toxicity, stability, and limited efficacy continue to represent considerable challenges [62,63,64,65,66]. Thus, sustained inhibition of miR_s_ and HIFs by therapeutic doses and a set schedule of SLM could lead to the activation of GPD-L1 and CPT1A and hyper-hydroxylation of PHD, resulting in the degradation of HIF_s_ and their regulated genes, including VEGF, GLUT1, and PD-L1. The role of miR_s_-210/-155 and HIF_s_ as critical therapeutic targets is well-established. The emphasis herein, however, is on the discovery that inhibition of these targets by SLM and MSC is dose- and schedule-dependent. Inhibition of these targets is necessary but not sufficient for achieving durable responses.

### 3.2. Transcription Factor Nrf2

Nuclear factor erythroid 2-related factor 2 (Nrf2) is a transcription factor that regulates the expression of an antioxidant response pathway, allowing cells to regulate reactive oxygen species and oxidative damage. Its activation promotes the antioxidant defense of normal cells. In normal tissues, Nrf2 protects tissues against oxidative damage [67,68,69,70,71,72,73,74,75,76]. In many cancers, however, the upregulated Nrf2 regulates growth and contributes to drug resistance [69,70]. Under oxidative stress, Nrf2 dissociates from KEAP1 and translocate into the nucleus, binding to the antioxidant response element and leading to the protection of cells from the oxidative stress induced by chemotherapy and radiation therapy. Nrf2 is regulated by multiple factors, including hypoxia, HIFs, miRs, cytokines, and TGF-ß, and regulates multiple target genes which promote tumor growth, metastasis, and drug resistance (Figure 3). Cellular expression of Nrf2 promotes the EMT [71]. The Nrf2 protein is positive in 119 out of 151 ccRCC tumors, compared to 87 out of 151 in normal tissue [69] (Table 3). It was demonstrated that selenium deficiency promotes the upregulation of Nrf2 [72,73,74,75]. We reported that a defined dose and schedule of MSC can differentially modulate the expression of Nrf2 in normal tissue vs. lung A549 and HT29 colon carcinoma cell lines and xenografts [72]. Treatment with MSC resulted in increased expression levels of Nrf2 in normal mouse tissues and decreased levels in tumor tissues. A schematic representation of the deferential effects of selenium in normal vs. tumor tissues was proposed, as outlined in Figure 4 [72]. Data generated in preclinical models demonstrated that treatment with a specific type, dose, and schedule of a selenium-containing molecule has dual effects on the expression levels of Nrf2 in tumor vs. normal tissues, downregulating its expression in tumors and upregulating it in normal tissues (Figure 4). These differential effects were associated with the selective sensitization of tumor tissues to subsequent treatment with chemotherapy. The documented protection of normal tissues from drug-induced toxicity by selenium may also be due, in part, to the activation of Nrf2, thus resulting in diminished levels of ROS and activation of PHDs, resulting in enhanced HIF hydroxylation and degradation. In brief, Nrf2 is a pleiotropic transcription factor that regulates multiple targets associated with increased tumor angiogenesis, tumor growth, mitochondrial metabolism, and drug response. For these reasons, it is potentially a critical druggable target. Hyperactivated Nrf2 plays dual roles, protecting normal tissues against oxidative damage and functioning in tumor tissue as an oncogenic protein. In tumor tissues, the Kelch-like ECH-associated protein (Keap1), an inhibitor of Nrf2, is inactivated, resulting in the stable expression of Nrf2, translocation to the nucleus, and activation of target genes. Based on preclinical data demonstrating that treatment with a defined dose and schedule of selenium resulted in the activation of Nrf2 in normal tissues and its downregulation in lung tumor tissue, it is expected that modulation of the dual function of Nrf2 by selenium will result in selective tumor tissue sensitization to subsequent treatment with immune checkpoint inhibitor-based immunotherapy alone and in combination with chemotherapy.

### 3.3. Transforming Growth Factor-Beta (TGF-β)

TGF-β, a multifunctional extracellular cytokine, acts as a tumor suppressor in normal tissues and as an immune-suppressor oncogene in advanced tumor tissues. It may be regulated differentially by the specific types of miRNAs, HIFs, and cytokines expressed in tumors and their associated microenvironments. TGF-β induces stabilization of HIFs under hypoxic and normoxic conditions by regulating biomarkers that regulate prolyhydroxylase-2 activity, namely glycogen synthase kinase 3 beta, which is overexpressed in 93% of ccRCC [77], glycerol-3-phosphate dehydrogenase-1 like, which is significantly expressed at lower levels in ccRCC than normal renal tissue [78], fumarate hydratase [79], succinate dehydrogenase [80], and pVHL [81]. TGF activity was attenuated by the reintroduction of VHL in ccRCC [81,82,83,84]. The altered expression of these biomarkers likely contributes to the instability of the tumor microenvironment that could result in decreased drug delivery to tumor cells. It was reported that TGF-β negatively regulates pericyte recruitment during blood vessel stabilization [85]. In addition, TGF-β regulates the expression of targets implicated in increased tumor angiogenesis and drug resistance, including programmed death-l and its ligand (PD-L/PD-L1), vascular endothelial growth factor (VEGF), and an immune response regulator, NKG2D [86]. NKG2D was also reported to be modulated by selenium [87]. TGF-β inhibits immune responses via the activation of FOXP3 and the regeneration of Treg. TGF-β is associated with increased Treg cells, IFN-g, decreased NK cytotoxicity, immune-suppressive tumor microenvironment, and increased tumor angiogenesis, due, in part, to the stable expression of PD-L1 and VEGF, as well as the induction of the EMT. Additionally, recent results demonstrated that, in patients with advanced COVID-19, TGF-β is overexpressed. Several studies have shown an association with blood selenium deficiency [88,89]. Patients with tumors overexpressing TGF-β in 76–100% of their cells demonstrated a median survival of 20 months, compared to 60 months for patients with only 0–25% of their cells overexpressing TGF-β. Evaluation of 32 renal cell carcinomas reported higher expression of FOXP3 expression levels in ccRCC compared with adjacent normal renal tissue [90]. Thus, TGF-β is overexpressed in the majority of advanced ccRCC tumors, is involved in the regulation of biomarkers implicated in drug resistance, increased tumor angiogenesis, and mitochondrial lipid metabolism, and is shown to be a target of selenium.

### 3.4. P-Glycoprotein (Pgp)

The multidrug-resistance glycoprotein (P-gp) is a member of the ABC transporter family and is an energy-dependent proton pump that facilitates the efflux of DNA-reactive drugs and protein kinase inhibitors from cancer cells, limiting the accumulation of therapeutically effective concentrations of drugs at target sites and rendering tumor cells resistant to structurally and mechanistically unrelated anticancer drugs. Pgp is expressed in ccRCC, and its expression is associated with hypoxia, HIFs, Nrf2, and specific types of microRNAs reported to modulate the expression of MDR1 mRNA and P-glycoprotein [41,91,92,93,94,95,96]. miRNAs interact directly with the 3′ untranslated region (UTR) of the MDR1 mRNA and suppress P-glycoprotein expression. Although the functional role of P-gp and other multidrug-resistant markers in resistance to cytotoxic drugs and radiation therapy has been documented, the potential role of P-gp in resistance to the anti-angiogenic TKIs and immune checkpoint inhibitors has not yet been fully elucidated. Unlike other human solid tumors, ccRCC is generally considered chemo/radiotherapy resistant. Recent data indicate that the biologically targeted molecules serve either as a substrate or as an inhibitor of P-gp, depending on the dose and treatment duration [41,92,93,94,95,96]. A study by Jedesko et al. demonstrates potentiation of the antitumor activity of pazopanib by topotecan, an inhibitor of HIF_s_ [97]. To date, however, agents developed to inhibit the functional role of Pgp are toxic and exhibit limited clinical therapeutic value. Due to the demonstration that HIFs, Nrf2, and miRs that regulate Pgp are selenium targets, it is reasonable to expect that selenium may directly or indirectly regulate the expression of Pgp, and that its effective and selective inhibition may result in the mitigation of resistance to biologically targeted molecules and chemotherapeutic drugs.

## 4. Modulators of the Proposed Druggable Targets

The current trend in the treatment of patients with advanced ccRCC is the continued development of clinical combinations of biologics that target the upregulated pro-angiogenic pathways with immune checkpoint inhibitors and mTOR inhibitors. The fact that most of these patients are nonresponsive, and that those who do respond eventually relapse, suggests the need to identify additional ‘druggable’ targets for those under current clinical development. miRs-210/-155, HIF-1α, HIF-2α, Nrf2, and TGF-β are highly expressed in the majority of advanced ccRCC tumors and cooperatively regulate the expression of biomarkers critical for the pathogenesis of ccRCC and potentially critical therapeutic targets. Recent clinical data demonstrated the potential benefit derived from the inhibition of HIF-2α [98,99,100]. Data generated suggests the possibility that while inhibition of a single target, HIF2α, may be necessary, it is not sufficient for achieving the necessary durable responses. Although significant efforts are underway to develop miRs and TGF-β inhibitors, toxicity and limited efficacy continue to represent major clinical challenges [101].

Two alternative approaches are presented as potential inhibitors of the indicated ‘druggable targets’, including: (1) the use of defined types and doses of multitarget selenium-containing molecules, and (2) topotecan as a transcriptional inhibitor of HIF_s_ and a topoisomerase 1 inhibitor. In addition, the overexpressed thymidine phosphorylase, a proangiogenic enzyme, offers the scientific rationale for its use as a substrate for activation of 5-Flourouracil pro-drugs to cytotoxic metabolites.

### 4.1. Selenium-Containing Molecules

Although several selenium-containing molecules are under preclinical and clinical evaluation, the focus of this mini-review is on Se-methyl selenocysteine (MSC) and seleno-L-methionine (SLM). Both agents are “pro-drugs” and are activated by β-lyase to the presumed active moiety, methylselenol [87]. These selenium compounds act as antioxidants, and therapeutic doses modulate multiple targets associated with increased tumor angiogenesis and drug resistance [102,103,104]. Using several human tumor xenograft models, it was demonstrated that miR_S_ -210/-155, HIF-1/2α, Nrf2, and regulated targets, PD-L1 and VEGF, are selenium targets, and their downregulation was associated with enhanced antitumor activity in multiple mouse models [53,54,55,56,57,105,106,107]. The proposed clinical use of specific types of selenium is not as a cytotoxic drug but rather as a selective modulator of the therapeutic efficacy and toxicity of anticancer therapeutics through targeting HIF_S_, miRs, and Nrf2, that regulate targets implicated in ‘drug’ resistance and increased tumor angiogenesis.

Selenium was evaluated in clinical prevention trials [108,109] and as a protector from induced chemotherapy and radiation toxicity [52,56,110,111,112,113]. Unlike the use of 200 mcg of SLM in the prevention trials, molecularly effective doses of SLM/MSC in sequential combination with anticancer drugs was synergistic in combination with anticancer drugs in several preclinical models. A recent laboratory-based clinical phase 1 trial (www.clinicaltrials.gov, NCT02535533, accessed on 1 December 2022), of up to a 4000 mcg total dose administered twice daily for fourteen days, then daily with axitinib, a tyrosine kinase inhibitor that targets VEGFRs, demonstrated significant efficacy in previously treated patients with advanced ccRCC [114]. The reported increased prostate cancer and diabetes in patients treated with selenium [107] was not seen in the 35 patients continuously treated with the high SLM dose for more than one year.

### 4.2. Topotecan

Topotecan is a topoisomerase 1 inhibitor and potent transcriptional inhibitor of HIFs that regulates multiple targets, including VEGF and PD-L1 [115,116,117,118]. The doses and schedules of topotecan clinically utilized have exhibited limited antitumor activity in ccRCC patients and are associated with significant dose-limiting toxicities. To reduce its toxicity and enhance its antitumor activity, metronomic use of topotecan has been evaluated in preclinical models, with promising outcomes [118]. We demonstrated that protracted administration of topotecan in combination with selenium was necessary for pronounced inhibition of HIF_S_ in ccRCC xenografts expressing a high intensity of HIFs and miR_S_-155 and -210. The protracted oral administration of this molecularly effective dose and schedule of topotecan in combination with MSC was not toxic and was synergistic in combination with biologic-targeted therapeutics [55]. The protracted administration of topotecan could exert dual effects that inhibit topoisomerase 1, leading to the induction of DNA double-strand breaks, a cytotoxic effect, while also inhibiting HIF_S_ synthesis, leading to inhibition of glycolytic enzymes required for anaerobic metabolism and of transcriptionally regulated proteins VEGF and PD-L1, an anti-angiogenic effect. Given the fact that over 80% of tumors from patients with advanced ccRCC express both HIF-1α and -2α [11,12], and with the knowledge that stable expression of HIFs is regulated by the rate of synthesis and degradation, it is expected that the combination of topotecan’s dose-sufficient inhibition of HIF_S_ synthesis with SLM that targets HIFs degradation will result in a more selective and efficacious treatment modality. Furthermore, since HIF_S_ regulate, in part, the stable expression of PD-L1 and VEGF, inhibition of HIF_S_ should sensitize tumor cells to treatment with anti-angiogenic and immune checkpoint inhibitors. The administration of a molecularly effective dose of topotecan, rather than the maximum tolerated dose, should result in less toxicity to host tissue with the potential for greater efficacy.

## 5. Thymidine Phosphorylase, an Activator of 5-Flourouracil Pro-Drugs

Thymidine phosphorylase (TP), also known as platelet-derived endothelial growth factor, is overexpressed in tumors of patients with ccRCC and other cancers [119,120] (Table 4). A study by Huang et al. demonstrated that in 127 ccRCC tumors, 54% expressed high levels of TP and 46% expressed low levels. Furthermore, the reported 5-year survival was 88.1% for high TP and 68.1% for low TP expression [118]. Therapeutically, the overexpression of TP in tumor tissues exerts dual effects: the promotion of tumor angiogenesis and metastasis, and an enzyme for the activation of several clinically approved 5-Flourouracil pro-drugs, such as capecitabine and S-1 [121,122]. Thus, utilization of TP for greater and selective activation of 5-FU pro-drugs could result in decreased tumor angiogenesis and a higher tumor level accumulation of 5-FU cytotoxic metabolites. Since pretreatment with therapeutically and molecularly effective doses of selenium resulted in stabilization of tumor vasculature [57,105], it is expected that higher drug concentrations will be delivered to tumor tissues, resulting in greater accumulation of 5-FU cytotoxic metabolites. Using ccRCC xenografts expressing TP, treatment with protracted low doses of capecitabine or S-1 were more active and selective than the schedule employed clinically, including the once-monthly or daily for 5 days every 21 days. With the demonstrated molecular and biological properties of selenium, the sequential combination of a defined dose of SLM in combination with capecitabine and targeted molecules should be clinically evaluated.

## 6. Discussion

The communication between tumor cells and the surrounding microenvironment contributes to ccRCC’s biological and molecular heterogeneity and resistance to standard therapies. The instability of tumor vasculature, a component of the TME, likely plays a critical role in limiting the delivery and distribution of treatment to tumor cells [3,39,123,124]. While the major focus of research has been on tumor cells, significant efforts are underway to characterize and understand the TME as a potential ‘druggable’ therapeutic target. While significant advances have been achieved, in the treatment of ccRCC patients, resistance and grade 3/4 dose-limiting toxicities [1] highlight the need to identify new ‘druggable’ targets and prioritize cancer immunotherapy. The proangiogenic and drug-resistant biomarkers, HIF-1α and -2α, miRs-210/-155, Nrf2, and TGF-β, are overexpressed in the majority of primary and metastatic ccRCC tumors. The FDA-approved belzutifan, an inhibitor of HIF-2α, is under clinical development, with promising efficacy in patients with advanced ccRCC [98,99,100]. ccRCC tumors, however, express both HIF-1α and -2α, which regulate independent and overlapping target genes [125,126,127]. Similarly, responses of patients with advanced ccRCC treated with immune checkpoint inhibitors were observed independent of PD-L1 expression. Thus, inhibition of a single target altered in a heterogeneous tumor and associated microenvironment is not sufficient for optimal treatment outcomes. Although the focus of this review is on specific druggable biomarkers altered in ccRCC tumors by a defined dose of specific types of selenium, advanced ccRCC tumors express multiple other molecular, immunologic, biologic, and metabolic biomarkers (Figure 1). ccRCC tumors are also characterized by cytoplasmic lipid accumulation [128,129] altered expression of a number of metabolic biomarkers [130,131,132,133,134,135], and nutritional status [136]. The unstable tumor microenvironment composed of multiple cell types plays a critical role in the regulation of tumor angiogenesis, metastasis, and the response to a variety of anticancer treatments [137,138]. The documented metabolic and molecular alteration associated with ccRCC tumors offers the potential for the development of drug inhibitors.

Using several xenograft models, it was demonstrated that specific types of selenium-containing molecules, Seleno-L-methionine (SLM) and Se-methyl-selenocysteine (MSC), are effective inhibitors of biomarkers overexpressed in the majority of ccRCC tumors, including HIF-1/2α, miR_S_-210/-155, Nrf2, and TGF-β. Downregulation of these altered biomarkers is associated with dual and complimentary effects, including the following: (1) normalizing tumor vasculature, resulting in enhanced drug delivery to tumor cells, and (2) increasing the time window for the administration of anticancer therapeutics. Therapeutic synergy was documented only when anticancer therapeutics were administered at a time during which optimal downregulation of these biomarkers and stabilization of tumor vasculature was achieved via 7–14 days of treatment with SLM or MSC. While ccRCC tumors are considered resistant to chemotherapy, the recognition that TP is overexpressed in ccRCC tumors provided the rationale for the use of capecitabine, a 5-FU pro-drug activated by TP, in combination with selenium and biologically targeted molecules. Utilization of TP as a therapeutic target could achieve dual effects, decreasing tumor angiogenesis and generating higher 5-FU cytotoxic metabolites, an approach that may make drug-resistant tumors sensitive.

Since SLM is FDA-approved clinically, we are introducing SLM, not as a cytotoxic molecule but as a pleiotropic modulator of multiple biomarkers altered in tumor tissues and their associated microenvironment, implicated in increased tumor angiogenesis, drug resistance, and immune evasion. SLM as multi-target modulators could avoid the use of individual target inhibitors and minimize associated toxicities. Preclinical results in several xenograft models demonstrated that durable responses were only achieved when the molecularly effective dose and schedule of SLM/MSC were administered sequentially in combination with chemotherapy and biologically targeted molecules. The therapeutic value of selenium in modulation of the efficacy of axitinib in previously treated patients with advanced ccRCC was demonstrated by the completed phase clinical trial [114]. Durable responses were documented in patients treated with an SLM dose that yielded blood selenium concentrations that were determined to be molecularly effective and therapeutically synergistic with anticancer drugs in xenografts. Delineation of the underlying mechanisms of action of therapeutic doses of SLM at the level of tumor cells and the associated tumor microenvironment could provide the scientific rationale for the development of more efficacious and selective mechanism-based combination therapies.

Recent research investigating the potential role of the nutritional status of patients infected with COVID-19 indicated that these patients are selenium-deficient, and that the selenium level is a factor associated with the severity of infection and morbidity [139,140]. In addition, with the knowledge that selenium targets immune response biomarkers, along with the data that we generated confirming that TGF-β is a selenium target by selenium concentrations that can be achieved clinically without host toxicity, as well as reports that TGF-β is overexpressed in patients with COVID-19 [88,141,142], sufficient evidence is available to evaluate the potential role of a defined selenium type and dose in the treatment of patients with COVID-19. Selenium supplementation alone and in combination with other treatments under evolution may reduce the onset of infection and may also reduce the severity of infection and morbidity.

## 7. Take Home Message

The intent of this minireview was not to provide a comprehensive review of all targets altered in ccRCC but to illustrate that specific targets overexpressed in ccRCC, such as HIF_S_, miRs-210/-155, TGF-β, and Nrf2, which are targets that can be downregulated by nontoxic, clinically achievable selenium doses. In addition, this minireview highlights that therapeutic synergy with a variety of anticancer drugs is associated with the downregulation of the intended targets that are implicated in multidrug resistance, an unstable tumor microenvironment, and increased tumor angiogenesis. The preclinical and clinical molecular effects and therapeutic benefits generated should be confirmed by other laboratories and should stimulate research to better understand the underlying mechanisms of action of molecularly effective and therapeutically efficacious doses of SLM or MSC. The SELECT prevention trial in prostate cancer that generated a negative impression as to the therapeutic value of selenium should not impede future research. Evidence for the overexpression of multidrug resistance molecules as selenium targets, topotecan as an inhibitor of HIF_S_ synthesis and a cytotoxic drug, together with the overexpression of thymidine phosphorylase, a proangiogenic and an enzyme essential for the activation of 5-Flourouracil pro-drugs, should provide the scientific rationale for the development of a combination of molecularly targeted molecules in combination with targeted chemotherapy.

## Figures and Tables

**Figure 1 ijms-24-00902-f001:**
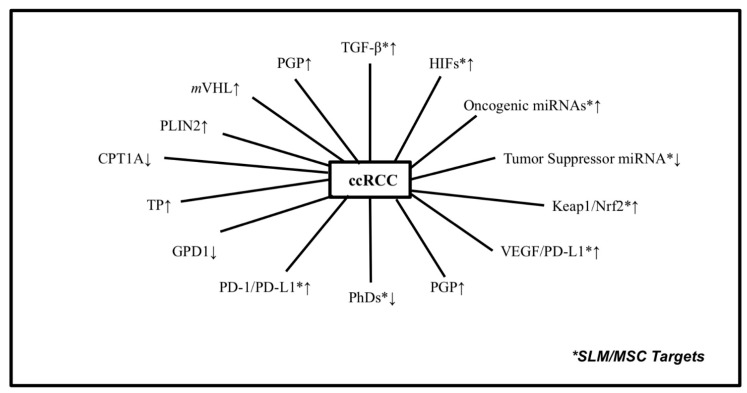
Molecular and immunologic targets expressed in ccRCC tumors and associated microenvironment implicated in tumor angiogenesis, unstable microenvironment, drug resistance, and immune evasion. TGF-ß, transforming growth factor-ß; HIFs, hypoxia-inducible factor1α/2α; onco-miRs, oncogenic microRNAs; TSmiR_S_, tumor-suppressor microRNA_S_; P-gp, p-glycoprotein; PHD, prolyl hydroxylase; PD-1/PD-L1, programmed death-1 and its ligand; GPDL1, glycerol-3-phosphate dehydrogenase 1-like; TP, thymidine phosphorylase; CPT1A, carnitine palmitoyl transferase IA; PLIN2, Perilipin 2; mVHL, mutated Von Hippel-Lindau tumor-suppressor gene; ↑ upregulated; ↓ downregulated. *: SLM/MSC targets.

**Figure 2 ijms-24-00902-f002:**
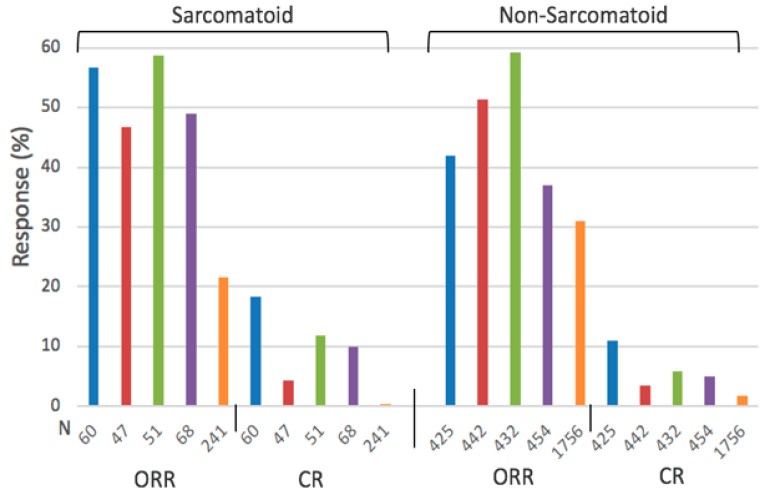
Objective response rate (ORR, (CR + PR)) and complete response (CR) of previously untreated patients with advanced ccRCC with and without sarcomatoid differentiation treated with a combination of biologically targeted molecules: randomized phase 3 clinical trial [1]. The number indicated on the *x*-axis represents the number of patients included in each clinical trial. Ipilimumab/Nivolumab (blue), Avelumab/axitinib (red), Pembrolizumab/Axitinib (green), Atezolizumab/Bevacizumab (purple), sunitinib (orange).

**Figure 3 ijms-24-00902-f003:**
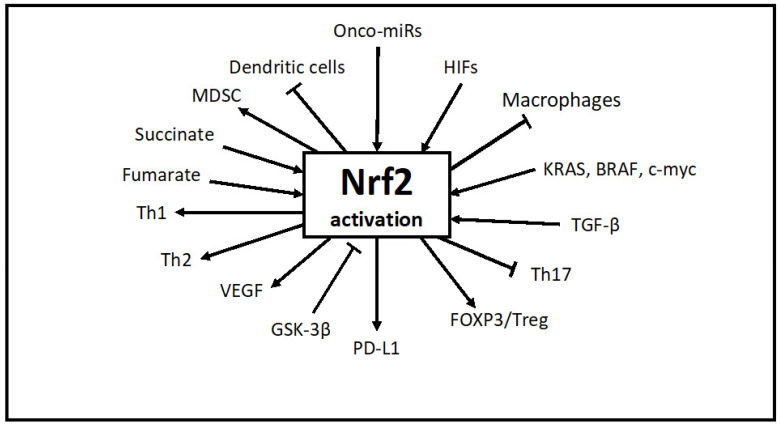
Biomarkers’ expression associated with the overexpressed nuclear factor erythroid 2-related factor 2 (Nrf2). ↑ Activator and inhibitor. Onco-miRs, oncogenic microRNAs; HIFs, hypoxia-inducible factor-1α/2α; KRAS, BRAF, c-myc, Kristen rat sarcoma virus, B-raf proto-oncogene, serine/threonine kinase, and c-myc is a proto-oncogene; TGF-ß, transforming growth factor-ß; Th17, T-helper 17 cells; Foxp3/Treg, transcription factor forkhead box protein/T regulatory cell; PD-L1, programmed death ligand 1; GSK-3ß, glycogen synthase kinase-3-beta; VEGF, vascular endothelial growth factor; Th1/2, T-helper-type 1/2 cells; MDSC, myeloid-derived suppressor cells [75].

**Figure 4 ijms-24-00902-f004:**
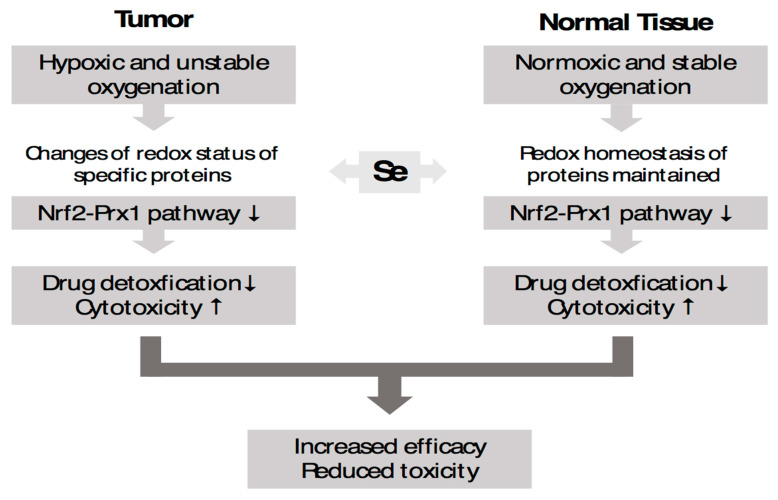
Differential regulation of Nrf2 expression by methylselenocystein (Se) in tumor and normal tissues [72]. Se upregulates Nrf2 in normal mouse tissues and downregulates its expression in tumor tissues.

**Table 2 ijms-24-00902-t002:** Summary of objective response rate (ORR), median progression-free survival (mPFS), and median overall survival (mOS) of ccRCC patients with and without sarcomatoid differentiation treated with a combination of biologically targeted molecules vs. sunitinib [1], ( ): range.

Outcome	Patients with Sarc. Differentiation		Patients without Sarc. Differentiation	
	Combo	Sunitinib	Combo	Sunitinib
ORR (%)	52.8 (46.8–58.8)	21.5 (14–31.5)	47.6 (37–59.8)	30.9 (25.7–35.7)
mPFS (Months)	7.9 (7.0–8.4)	5.7 (4.0–8.3)	12.1 (8.2–15.1)	9.1 (8.3–11.1)
mOS (Months)	24.8 (18.3, 31.2)	14.3 (13.6, 18.3)	34.6 (33.6, 35.6)	30.8 (26.6, 34.9)

**Table 3 ijms-24-00902-t003:** Expression of Nrf2 in ccRCC tumors and normal tissue [69]. Difference in the expression levels between tumor and normal tissue is significant, with *p* < 0.001.

Site	n	Positive Nrf2 Expression
ccRCC tumor	152	119 (78.3%)
Normal tissue	151	87 (57.6%)

**Table 4 ijms-24-00902-t004:** Thymidine phosphorylase (TP) is preferentially expressed at a higher activity level in renal cell carcinoma (RCC) tumor cells than non-neoplastic kidney tissue specimens [119], range: ( ). Difference in the activity of TP between RCC and kidney is significant, with *p* < 0.0001.

Tissue	n	Median TP Activity (u/mg Protein)
RCC	116	12.8 (3.2–933.9)
Kidney	90	11.79 (0–128.0)

## Data Availability

Not applicable.

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
