# Peer review of "Druggable Biomarkers Altered in Clear Cell Renal Cell Carcinoma: Strategy for the Development of Mechanism-Based Combination Therapy"

_ijms, 2023, doi:10.3390/ijms24020902_

Round 1
Reviewer 1 Report
The manuscript titled “Druggable biomarkers altered in clear cell renal cell carcinoma: Strategy for the development of mechanism-based combination therapy” focuses on clear cell renal cell carcinoma to summarize the signaling and its therapy. The followings are some concerns and comments have been pointed out that the authors may want to consider.
1) Line 12: The abstract is too long. Please revise.
2) Lines 48-70: The authors cited 78 references in a short paragraph. Server sentences have over ten or even 21 references. But the cited articles did not well extended in the following review. Lots of statements were without proper citations. I’d highly suggest the authors pick the necessary papers and cite them properly. Only in this way the readers can easily track them.
3) Line 72: Please provide a high-resolution image.
4) Line 75 Table 1: Why does the reference number jump to 277?
5) Line 86 Figure 2:a) Please include sample sizes. b) Please provide a high-resolution image.
6) Line 89 Table 2: Please provide a high-resolution image.
7) Lines 105-107: Please delete unnecessary self-citations or cited them properly instead of stacking them all together.
8) Line 286: Please use a high-resolution image.
9) Line 287 Table 3: Please use italic p as it refers to a p-value throughout the manuscript.
10) Line 290: Please provide a high-resolution image.
11) Lines 395-407: Please modify the font to match with others.
12) Line 465: Please provide a high-resolution image.
13) Please seriously deal with the citation and format throughout the manuscript.
14) I’d highly suggest the authors remove too many repeated statements “our laboratory was the first…” throughout the manuscript.
Reviewer 2 Report
The review "Druggable biomarkers altered in clear cell renal cell carcinoma: Strategy for the development of mechanism-based combination therapy ", presented by Youcef Rustum et al., presents the main targetable biomarkers in ccRCC and also describes the potential role of the antioxidant seleno L methionine (SLM) as therapeutic drug in combinatorial therapies.
Minor points:
1- Quality of tables and figures is extremely poor. They must be improved to be published.
2- Table and figure legends lack a complete explanation of the message shown and the abbreviations.
Major points:
3- I miss an explanation of the fails recorded in trials using drugs such as sunitinib or bevacizumab as monotherapies.
I also miss a complete introduction and description of other promising molecular in ccRCC.
4- Finally, for a review, I find this manuscript as a summary of the results from a single lab, and the enumeration of the obtained results and potential therapeutical properties of SLM in ccRCC.
Therefore, I cannot accept this review in the present form.
Round 2
Reviewer 1 Report
Thank you for the update. Please find the comments below.
1: Line 12: No conclusion in the abstract.
2: Line 24: Please check “our laboratory” throughout the manuscript. I’d highly suggest the authors modify this description.
3: Line 59: There are some messed up symbols in Figure 1. Please fix them and provide the necessary abbreviations.
4: Line 87: a) Figure 2 is not clear. Please provide a high-resolution image. b) The reference should be cited properly.
5: Line 92 Table 2: a) “0” is not a letter for the word “objective”. b) what’s the meaning of (1)? Please cite references properly according to the journal guidelines to the authors.
6: Line 225 Figure 3: At least a general figure legend is still needed.
7: Line 231 Table 3: Please use italic p as it refers to a p-value throughout the manuscript.
8: Line 194: Please use β instead of b. Check throughout the manuscript.
9: Line 322: The letter “s” for “HIFs” should be lowercase. Check throughout the manuscript.
10: Please be consistent with the formats throughout the manuscript. Check them carefully. For example, “miR”, “miRNA”, and so on. There are lots of modifications need to do.
Reviewer 2 Report
The second version of the review "Druggable biomarkers altered in clear cell renal cell carcinoma: Strategy for the development of mechanism-based combination therapy ", presented by Youcef Rustum et al., has improved much more and I congratulate for this.
As I said before, it presents the main targetable biomarkers in ccRCC and also describes the potential role of the antioxidant seleno L methionine (SLM) as therapeutic drug in combinatorial therapies.
1- Abstract has been shortened, making it much more readable. I miss a final sentence or conclusion.
2- Figure 1 lacks the “N” number of evaluable patients included in each trial. There’s an arrow in between the CPT1A box.
3- Figure 2; quality must be improved. Remove the reference on the figure and explain/describe the (drugs) abbreviations.
4- Figure 4- Indicate what means “70”, please.
5- Line 85 - beta symbol is missing.
6- Line 339 - complete or fill the brackets in Tables 2 and 4.
7- Line 377 - for “aggressiveness” you mean “tumor malignancy”? Please, clarify.
8- Line 656 (figure 3)- Final closing bracket.
9- Line 761 - Include the abbreviation EMT instead of epithelial mesenchymal transition.
10- Lines 771-774. “…we have evidence to support the expectation that treatment with a defined selenium type, dose, and schedule may provide a broad approach to augment immune response in cancer patients and in patients with COVID-19.” Even that it is explained much later in a paragraph (1222-1232), please, refer or remove the sentence.
11- Line 935. Is 400 ug the total administered amount? Please, clarify.
12- Line 938- Capital after dot in “recent”
13- Line 954 – A reference is missed.
14- Lines 1033-1035 – Sentence unclear.
15- Sentence 1037-1040. “Because pretreatment with therapeutically and molecularly effective doses of selenium resulted in stabilization of tumor vasculature, (56,105), it is expected that higher drug concentrations will be delivered to tumor tissues, resulting in greater accumulation of 5-FU cytotoxic metabolites.” Based on the commented along the review, seems that selenium and its derivatives are not toxic but, maybe I’m wrong, there is no reference or data indicating such a thing. Please refer it using data based on human samples/trials.
Indicate the reference for the clinical trials cited (NTC#...)
16- I suggest a final grammar English edition since the huge number of changes make a bit difficult to rate. Nevertheless, I’m confident that the English will need very few changes.
17- I miss an explanation of the fails recorded in trials using drugs such as sunitinib or bevacizumab as monotherapies. I also miss a short description of other promising molecules in ccRCC.
18- I still find in this review a slight bias on self-mentions; personally, I don’t like it, it’ my own appreciation. Therefore, I led the decision to reduce or remove then to the editorial team of IJMS.
